# Perception and Coverage of Conventional Vaccination among University Students from Rouen (Normandy), France in 2021

**DOI:** 10.3390/vaccines10060908

**Published:** 2022-06-07

**Authors:** Andreina Arias, Joel Ladner, Marie-Pierre Tavolacci

**Affiliations:** 1Department of Epidemiology and Health Promotion, CHU Rouen, Inserm U 1073, Université de Rouen Normandie, F-76000 Rouen, France; ariass.andreina@gmail.com (A.A.); joel.ladner@chu-rouen.fr (J.L.); 2CIC-CRB 1404, CHU Rouen, Inserm U 1073, Université de Rouen Normandie, F-76000 Rouen, France

**Keywords:** vaccination coverage, university students, confidency, utility

## Abstract

The aim of this study was to assess vaccination perception and the prevalence of the overall vaccination coverage (VC) and associated factors among university students. An online study was conducted among students of a university in Rouen (Normandy), France, in January 2021, with questions about the VC and perception of the vaccines. The convenience sample included 3089 students (response rate of 10.3%), with a mean age of 20.3 (SD = 1.9). The overall VC was 27.8% (39.2% for the healthcare students (HCS) and 21.3% for the non-HCS; *p* < 0.0001). Confidence (efficacy and security) was lower than the conviction of usefulness. The characteristics associated with VC were having the intention to be vaccinated against COVID-19, high perceptions of usefulness for their own health, having confidence in the vaccines’ efficacy and security, and a high estimated level of knowledge about vaccination. Education about the general interest and mechanism of action of vaccines could improve the perception of vaccines. Then, it is relevant to improve vaccination literacy and confidence in university students, who, as future adults and parents, will vaccinate themselves and their children; as well as healthcare students who are future healthcare workers and, therefore, will vaccinate and counsel their patients.

## 1. Introduction

Vaccination, which is one of the most cost-effective public health interventions in combating infectious diseases [1], has substantially reduced the number of infectious disease cases and prevented an extensive number of cancer diseases, disability and mortality worldwide. Nevertheless, for a few decades, despite years of accumulative scientific evidence that support the effectiveness of vaccines, vaccination has been perceived as unsafe and unnecessary by a growing number of individuals in the world, especially in most Western countries [2]. The concept of “vaccine hesitancy” means to delay accepting or refusing vaccination despite vaccination services being available. Vaccine hesitancy is considered one of the top 10 threats to world health [3] and has also been steadily increasing in more than 90% of countries [4], which can lead to uptake rates of certain vaccines being suboptimal or unsatisfactory. Acceptance of vaccination is a behavior outcome resulting from a complex decision-making process that can be potentially influenced by a wide range of factors. Betsch et al. developed the “5C” psychological construct to understand the psychological underpinnings of vaccine uptake: “confidence”, “complacency”, “constraints”, “calculation”, and “collective responsibility” [5]. The health belief model is one of the most widely used models for examining the relationship between health behavior and the use of health services and could be used to better understand the compliance of vaccines [6]. The French vaccination coverage goal is set to reach 95% (except for seasonal influenza). There is no routine collection of vaccine coverage (VC) in France and little is known about vaccination in young adults [7]. The meningococcal C vaccine has the lowest coverage in France among 15–24-year-olds [8], and the Human Papillomavirus (HPV) vaccine has a vaccine coverage below target [9]. HPV vaccination has been offered to 11-year-old girls since 2007 and was introduced for 11-year-old males in France in 2020. Studies about vaccination coverage are sparse among students, and they mainly concern healthcare students (HCS) [10]. Meningococcal and HPV vaccines could be requested by the student if they have not been given during adolescence. Indeed, HCS are future healthcare workers and will vaccinate and counsel their patients [11]; the conviction and motivation of the healthcare worker could mitigate vaccine hesitancy in patients. It is relevant, therefore, to not only study VC among HCS but also in the other curricula because these students are also future adults and parents who will vaccinate themselves and their children. Acceptance of conventional vaccination also implies acceptance of a new vaccine in times of a pandemic, such as the COVID-19 vaccine, which will be implemented in 2021 [12]. Galle et al. showed that COVID-19 vaccination acceptance was found to be related to having a previous vaccination against influenza [13]. COVID-19 vaccine hesitancy in college students correlated strongly with some concerns about vaccines in general [14]. Therefore, it is important to identify the VC of university students and related factors to design and evaluate public health strategies beyond the current pandemic. The aim of this study was to assess vaccination perception (attitudes and beliefs, the prevalence of each vaccine (except COVID-19)) and the associated factors of overall VC among healthcare and non-healthcare students.

## 2. Methods

### 2.1. Study Design and Settings

A cross-sectional study related to “Ta Santé en un Clic” was conducted among 30,000 students at the University of Rouen-Normandy, France. The survey was comprised of an original questionnaire written in the French language. The electronic questionnaire was sent via the students’ university e-mail list to the students of Rouen-Normandy University with two e-mail reminders from the 7–31 January 2021. Volunteer students filled in an anonymous online questionnaire. The Rouen University Hospital’s Institutional Review Board, without mandatory informed consent (E-2021-01), approved the observational study design according to the Helsinki declaration.

### 2.2. Data Collection

#### Sociodemographic Data

The data collected were gender, age, the year level of study, and course studied and were classified into two categories: healthcare and non-healthcare students.

### 2.3. Vaccination Perception and Knowledge

The health belief model (engagement beliefs about health problems, perceived benefits of action and barriers to action, and self-efficacy) was used for [6]. The perception of general vaccination (excluding vaccines against COVID-19) was collected with these questions: “Do you think that getting vaccinated is useful for your health?”, “Do you think that getting vaccinated is useful for the health of others?”, “Do you trust in the efficacy of the vaccines?”, and “Do you trust in the security of the vaccines?” These questions were scaled from 0 to 10 (0 being not useful at all/not at all confident to 10 being very useful/very confident). Further, the survey questioned their self-perceived level of knowledge about vaccination with the question: “How do you estimate your level of knowledge about vaccination (vaccination schedule, recommendations, interest…)?” with a scale from 0 to 10 (0 being “I don’t know anything at all” and 10 being “I know very well”).

### 2.4. Vaccination Coverage Status

The questions were: “Do you have a vaccination notebook (“Yes”, “No” or “I don’t know”) and the status of vaccination: diphtheria, tetanus, pertussis, poliomyelitis (DTaP/IPV), measles, mumps and rubella (MMR), BCG (Bacillus Calmette-Guérin vaccine), human papillomavirus (HPV) for female students, hepatitis B virus (HBV), hepatitis A virus (HAV), meningococcus C, and the flu for the 2020–2021 season regarding the French 2019 guidelines for vaccinations [15]. The answers were: “Vaccinated and up to date with booster”, “Vaccinated but not up to date with booster”, “Not vaccinated”, and “I don’t know” except for the vaccines for the meningococcal C and the seasonal flu, for which the options answers were: “Vaccinated”, “Not vaccinated”, and “I don’t know”. Eleven vaccines became mandatory in 2018: DTaP/IPV, MMR, HBV, meningococcus C, pneumococcus and Haemophilus [15]. An overall VC was defined when these 11 mandatory vaccines were up to date in 2018. Students were asked about their vaccination intentions when the COVID-19 vaccine becomes available.

### 2.5. Statistical Analysis

Continuous variables were expressed as means and standard deviation (SD) compared with the Student’s t-test, and discrete variables were reported as percentages compared with the Chi-Square (χ²) test. The effect size with Cohen’s d was calculated for the continuous variables. All factors with *p*-values lower than 0.25 were integrated into the multivariate logistic regression model. The adjusted odds ratio (AOR) and 95% confidence intervals (CI) were calculated. The answers were mandatory, then there was no missing data.

## 3. Results

### 3.1. Population Characteristics

A total of 3089 students were included in this study (participation rate of 10.3%), with a mean age of 20.3 (SD = 1.9). Overall, 71.4% of participants were women and 38.8% were healthcare students. Out of the 3089 students, 92.8% stated that they had their own vaccination notebook and 4.4% did not know. The characteristics of the population are presented in Table 1.

### 3.2. Vaccination Perception

The perception of usefulness for personal health and others’ health, confidence in the efficacy and the security of the vaccination, and knowledge about the vaccination according to the curriculum, are displayed in Figure 1. Students’ interest in vaccination was no different for personal health (mean 8.6; SD:2.03) than others’ health (mean 8.5; SD:2.30) (*p* = 0.30), and they had more confidence in the effectiveness (mean 8.0; SD (2.27)) than safety (mean 7.7 (SD: 2.34)); *p* < 0.0001. Declared knowledge about vaccination was low with a mean of 5.8 (standard deviation (SD) = 2.3) and a higher level of knowledge among HCS that non-HCS, respectively, 7.0 (SD = 1.8) and 5.1 (SD = 2.4); *p* < 0.0001.

### 3.3. Vaccination Coverage

The VC, according to the vaccine, are displayed in Figure 2. The highest vaccination rate was for the mandatory vaccination (DTP pertussis): 94.1% for HCS and 76.2% for non-HCS students (*p* < 0.0001). The overall VC was 27.8%, with a 95% confidence interval (26.3–29.4), (39.2% for HCS and 21.3% for non-HCS; *p* < 0.0001). There was no difference in the overall VC between men (27.0%) and women (28.5%) *p* = 0.22. The characteristics of students according to the overall VC status are displayed in Table 1. After logistic regression, being a woman, HCS, and having intentions to be vaccinated against COVID-19, as well as having high perceptions of its usefulness for one’s own health, confidence in the efficacy and security, and a high estimated level of knowledge about vaccination, were positively associated factors to being overall vaccinated. Being in the first year of the curriculum was associated with a lower overall VC than being in the fourth year and above.

## 4. Discussion

To our knowledge, our study is the first to establish a relationship between the vaccination coverage of university students and the quality of their perception of the usefulness, efficacy and safety of vaccination. A quarter to a third of non-HCS were up to date on all 11 vaccines currently required [15]. The mandatory vaccine (DTaP/IPV) has the highest VC (94.1% among HCS and 76.3% among non-HCS). In our study, two-thirds of female HCS and half of non-HCS are being vaccinated against HPV, which is higher than among female HCS in Italy (40%) [16]. Although college may be an opportune time to reach young adults for HPV vaccination, obstacles, including navigating parental influence, independent decision-making, and lack of awareness of vaccination status may impede vaccination during this time [17]. The perceived risk of HPV has emerged as a significant mediator in the uptake of the HPV vaccine in female college students [18]; a study in France showed different perceptions of the risks and benefits between mother and girl [19]. Wilson et al. showed that being confident in their ability to initiate and guide vaccine conversations with patients and parents ensures a high uptake of the HPV vaccine in the future in the general population [20]. The meningococcal VC was higher than in the general population (25%) [9]. VC of Hemophilus influenzae is not mandatory for HCS students but indicates an involvement in prevention for at-risk populations and is associated with a vaccine intention for a new vaccine in the epidemic period COVID-19 [21].

Whatever the vaccine, coverage is better among HCS than non-HCS. This is new knowledge because previous studies only involved HCS [10]. This could be explained by the systematic check of vaccines by the university’s healthcare service before their internships in the hospital: Hepatitis B was mandatory and the flu vaccination was recommended for health students. Moreover, HCS may be more aware of vaccination through their courses. Knowledge about conventional vaccination is better among HCS students than non-HCS students, as is also reported for knowledge about the COVID-19 vaccination [13]. In a time of epidemics, vaccination of healthcare workers is a key measure in the prevention of healthcare-associated infections due to their close contact with these populations of high-risk patients [22]. Furthermore, outside of the epidemic period, the influence of vaccination acceptance among students (e.g., HCS) should not be neglected, as they could possibly play the role of ambassador to their peer population of university students and have a better perception throughout their healthcare career [23,24]. In France, a new program of primary prevention interventions among HCS started in 2019, called ‘‘Service Sanitaire des Etudiants en Santé”, and has been shown to improve misconceptions and hesitancy surrounding vaccines [25]. Our study showed that female students have better VC than male students, while Bajos et al. highlighted a gendered reluctance toward vaccination in general (but even more so, regarding vaccination against COVID-19) [26]. This could be explained by better follow-up by a healthcare professional due to the prescription of a contraceptive method [27]. Young students in the first year of the curriculum had the lowest overall VC, which has previously been found in HCS [28].

We also highlighted that good VC is also associated with an intention to use a new vaccine, such as for the COVID-19 pandemic; hence, the importance of reinforcing adherence to traditional vaccination to prepare individuals for situations that require a rapid reaction, such as during a pandemic [29,30].

Confidence (efficacy and security) was lower than the conviction of usefulness, also reported in the European general population [31]. Collective responsibility is not involved in VC, whereas an interest in one’s health increases VC. In our study, low vaccine knowledge is associated with the lowest VC; hence, the importance of raising awareness among all students [32]. Furthermore, it could be relevant to improving vaccine literacy among university students [33]. Vaccine literacy has been built on the same idea as health literacy: it has been defined as a process of providing vaccine information, building communication, and increasing people’s engagement with vaccines [34,35]. Digital health literacy [36] and digital gamification [37] could be pathways to vaccine literacy for university students.

The quickest factor to implement and increase vaccine acceptance, as advised by the WHO, is to adopt three strategies: harness social influences (medical students could be an especially strong influence as peer students); increase motivation (through open and transparent dialogue and communication about the uncertainty and risks, including around the safety and benefits of vaccination); and create an enabling environment (making vaccination easy, quick and affordable) [38]. Having vaccine spokespeople who are trusted by vaccine-hesitant social groups is also beneficial [39,40].

## 5. Limitations

The student population in the city of Rouen, France, is made up of approximately 30,000 individuals, and the response rate in this study was around 10%, which could engender a response bias and may limit its generalizability and representativeness. The proportion of women is slightly higher than women students in France (60%) [41]. The proportion of HCS was higher in our study (39%) than in the University of Rouen-Normandy (15%); however, HCS and non-HCS were separately analyzed, which limited the selection bias. The extrapolation of results should be taken with caution because of the heterogeneity by demographic factors as the country in the respondents’ willingness to accept a vaccine [42]. The data on vaccination status were self-reported, which could engender information bias due to the fact that the information collected has to rely on the respondents’ memories of their past vaccines; it is not possible to verify a vaccination status or confirm the possession of a vaccination card by each student included in the sample.

## 6. Conclusions

Students perceive vaccines as helpful to their own health, however, are less confident in their safety. Despite having a positive perception about the benefits of vaccination, the vaccination coverage of students is not close to 100% when regarding the vaccines of the mandatory scheme. Healthcare students present the best perception, in general, about the usefulness of vaccination, as well as having better confidence in its efficacy and greater knowledge of the interest in vaccination, which could be correlated with their better vaccination rates compared to students in other curricula. Finally, our results suggest that better education about the general interest and mechanism of action of vaccines, could improve the perception of vaccine safety and the status of vaccination coverage in university students, and thus, their future children and patients, specifically for HCS eHealth and media literacies, where there should be a university skill to empower university students about vaccination and facilitate emergency vaccination in the event of a future epidemic [43,44]. Further research should examine VCs of universities, now that students and the population are vaccinated for COVID-19.

## Figures and Tables

**Figure 1 vaccines-10-00908-f001:**
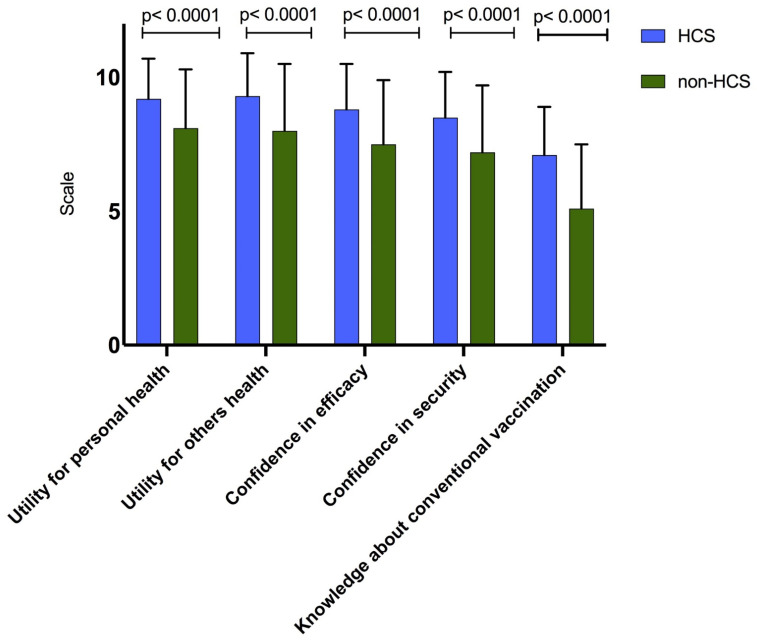
Utility, confidence and knowledge in vaccines among university students in France (*n* = 3089). HCS: Healthcare students.

**Figure 2 vaccines-10-00908-f002:**
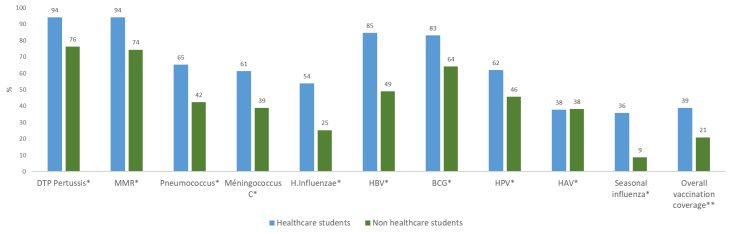
Vaccination coverage among university students in France (*n* = 3089). *: *p* < 0.05. ** Overall VC: Vaccination with DTPcoq, MMR, HBV, meningococcus C, pneumococcus, Haemophilus.

**Table 1 vaccines-10-00908-t001:** Overall vaccination coverage and associated factors among university students (France) *n* = 3089.

	Overall Vaccination Coverage (*n* = 860)	Non Overall Vaccination Coverage (*n* = 2229)	*p*	Effect Size d	Total (*n* = 3089)	Overall Vaccination Coverage AOR (95% CI)
Men (%)	27.0	29.2	0.22		28.6	Ref
Women (%)	73.0	70.8			71.4	1.23 (1.02–1.48)
Years of graduate school			<0.0001			
1st year (%)	25.6	34.7			32.2	0.76 (0.60–0.95)
2nd and 3th year (%)	43.4	44.4			44.2	0.76 (0.70–1.05)
4th year and upper (%)	31.0	20.8			23.6	Ref
Curriculum			<0.0001			
Non-healthcare students (%)	45.9	67.1			61.2	Ref
Healthcare students (%)	54.1	32.8			38.8	1.43 (1.19–1.72)
Useful for the personal health * mean (SD)	9.2 (1.63)	8.3 (2.11)	<0.0001	0.47	8.6 (2.03)	1.10 (1.01–1.19)
Useful for the others health * mean (SD)	9.2 (1.84)	8.3 (2.40)	<0.0001	0.43	8.5 (2.30)	0.99 (0.93–1.06)
Confidence about the vaccination efficacy * mean (SD)	8.7 (1.83)	7.7 (2.40)	<0.0001	0.46	8.0 (2.27)	0.97 (0.90–1.05)
Confidence about the vaccination security * mean (SD)	8.6 (1.93)	7.4 (2.41)	<0.0001	0.51	7.7 (2.34)	1.13 (1.07–1.70)
Knowledge about vaccination mean (SD)	6.7 (2.17)	5.6 (2.35)	<0.0001	0.50	5.9 (2.35)	1.12 (1.07–1.17)
Intention to vaccine against COVID-19 (%)	73.8	51.8	<0.0001		57.9	1.37 (1.11–1.70)

* except COVID-19 vaccination. Overall VC: Vaccination with DTPcoq, MMR, HBV, meningococcus C, pneumococcus, Haemophilus.

## Data Availability

The authors are grateful for the support of Joel Alexandre, President of Rouen Normandie University and Benoit Veber, Dean of the healthcare faculty of Rouen Normandie University.

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
