# Peer review of "Perception and Coverage of Conventional Vaccination among University Students from Rouen (Normandy), France in 2021"

_vaccines, 2022, doi:10.3390/vaccines10060908_

Round 1

Reviewer 1 Report

Thank you for the opportunity to review this manuscript. The study aimed to examine vaccine coverage and perception among students attending a university in France. The study findings can help identify where education about vaccines is needed.  The manuscript would be strengthened through more detail about the context of the study and analysis methods. My specific comments are below.

major comments

The term “the vaccination” used in the title and abstract made me think this paper was all about the the Covid-19 vaccine. Consider rewording for clarity (eg remove “the”) and providing some context in the abstract that you are considering overall vaccination coverage.

I find the introduction hard to follow as it does not include paragraphs and jumps around between points. Consider including more information about the schedule and coverage of the vaccines asked about in the survey. It is not clear why HPV and menigoccocal C vaccines are highlighted. Are there other studies in France about vaccine coverage? Given the timing of this study I think some context about the COVID-19 vaccines is also needed. The study appears to be conducted prior to the release of COVID-19 vaccines so this would be important information to include.

The discussion would also be improved by the use of paragraphs. For example a paragraph with the text about HPV as a separate paragraph and another about HCS. At present it is hard to follow. Conducting the study at 1 university is also a limitation as the results may not be generalisable to other areas and should be included in the strengths and limitations section.

More detail is required in the methods:

In section 2.1 more detail is needed about the survey -  How long did the survey take, was a reminder sent, was the survey in French?

On page 2 line 68 it states that Covid-19 vaccines were excluded. How was this done? Was there any information given to participants about which vaccines to consider – they may feel differently about different vaccines.

In section 2.5 on statistical analysis, in the current text it is not clear what is being compared in the statistical tests or which factors were tested for inclusion in the logistic regression? It should also be explained how global vaccination coverage is defined.

More detail is also needed in the results section:

Page 3 line 109: Knowledge is not included in Figure 1 as stated in the text.

Page 3 line 110-112: Were the differences mentioned between personal health and others health and effectiveness and safety significant? Include p value for these tests.

Table 1: Include p value for multiple logistic regression. Men should also be included in the table as the reference category.

Minor comments

Abstract

Page 1 Line 11: Explain what abbreviation VC is the first time appears

Page 1 Line 16: What is meant by score confidence?

Page 1 Line 18: “the own’s health” change to “their own health”

Page 1 Line 19-22: incomplete sentence “education about the general interest and mechanism of action of vaccines”

Page 1 Line 20-22: revise sentence to improve clarity.

Introduction

Eg Page 1 line 26: “diseases cases” to “disease cases”

Page 1 line 38: missing ] – check throughout as others missing as well.

Page 1 line 38: Does national here refer to France. Consider introducing France earlier.

Page 1 line 40: “little is known about young adults” change to “little is know about vaccination in young adults”.

Page 1 line 41-2: include what the coverage is for these vaccines.

Page 1 line 44: studies about what “are sparse among students”?

Methods

Page 2 section 2.1: Include more details about the survey - How long did the survey take, was a reminder sent, was the survey in French?

Page 2 line 68: How were Covid-19 vaccines excluded? Was there any information given to participants about which vaccines to consider – they may feel differently about different vaccines eg measles compared to HPV.

Page 2 line 81: In the introduction it states that HPV vaccines is now offered to 11 year old males why was the question about HPV VC only asked of females?

Page 2 line 87: it is not clear what the global VC refers to. Is this a question in the survey?

Page 2 line 88: expand what DTPcoq is

Page 2 line 93: “student test” change to “student t-test”

Page 2 line 93: “qualitative variables” I think should be discrete variables

Page 2 section 2.5: it is not clear what is being compared in the statistical tests. Which factors were tested for inclusion in the logistic regression?

Page 2 section 2.5: How is global vaccination coverage defined? Is it answering yes to having all listed vaccines and boosters?

Results

Page 3 line 100: what is consent rate?

Page 3 line 100: “mean of age” change to “mean age”

Page 3 line 101: “with a mean of age of 20.3 100 (SD=1.9), 71.4% of women and 38.8% of healthcare students.” change to “with a mean of age of 20.3 100 (SD=1.9). Overall 71.4% of participants were women and 38.8% were healthcare students.”

Page 3 line 110-112: Include the values in the text eg “Students' interest in vaccination is higher for personal health (mean 8.6, SD 2.03)”.

Figure 1: remove “the” from all axis categories eg “confidence in efficacy” change to “confidence in efficacy

Discussion

Page 5 line 148: “cursus”should this be “courses”?

Author Response

Thank you for the opportunity to review this manuscript. The study aimed to examine vaccine coverage and perception among students attending a university in France. The study findings can help identify where education about vaccines is needed.  The manuscript would be strengthened through more detail about the context of the study and analysis methods. My specific comments are below.

major comments

We thank the reviewer for their helpful comments to improve our manuscript.

The term “the vaccination” used in the title and abstract made me think this paper was all about the the Covid-19 vaccine. Consider rewording for clarity (eg remove “the”) and providing some context in the abstract that you are considering overall vaccination coverage.

We remove « the » and add « conventional » to indicate that it is not the COVID-19 vaccination

« Gloval vaccination coverage » has been changed to « overall vaccination coverage »

I find the introduction hard to follow as it does not include paragraphs and jumps around between points. Consider including more information about the schedule and coverage of the vaccines asked about in the survey. It is not clear why HPV and menigoccocal C vaccines are highlighted. Are there other studies in France about vaccine coverage? Given the timing of this study I think some context about the COVID-19 vaccines is also needed. The study appears to be conducted prior to the release of COVID-19 vaccines so this would be important information to include.

We hihglighted the Meningococcal and HPV vaccine because these 2 vaccines are not necessarily done at the request of parents but could be also by the students. « Meningococcal and HPV vaccines could.  can be requested by the student if they have not been done during adolescence »

 The only study on student vaccination coverage in France is by Baldolli  et al and this study was only in healtcare students

As sugested we have specified that the study was carried out just before the introduction of the COVID-19 vaccination. Acceptance of conventional vaccination also implies acceptance of a new vaccine in times of a pandemic, such as COVID-19 vaccine that will be implemented in 2021

The discussion would also be improved by the use of paragraphs. For example a paragraph with the text about HPV as a separate paragraph and another about HCS. At present it is hard to follow. Conducting the study at 1 university is also a limitation as the results may not be generalisable to other areas and should be included in the strengths and limitations section.

As suggested, paragraphs have been done

Conducting the study at a single university is a limitation  that we indicated in the limitation section »The student population in the city of Rouen, France, is made up of approximately 30,000 individuals, and the response rate in this study was around 10%, which could engender a response bias and may limit its generalizability and representativeness.  The proportion of women is slighty hiher than women students in France (60%). The proportion of HCS was higher in our study (39%) than in the University of Rouen-Normandy (15%); however, HCS and non-HCS were separately analyzed, which limited the selection bias. « 

More detail is required in the methods:

In section 2.1 more detail is needed about the survey -  How long did the survey take, was a reminder sent, was the survey in French?

We specifed :’ The electronic questionnaire was send via the university mailing list of the students of Rouen-Normandy University with two e-mail reminder from the 7th to the 31st of January 2021’

On page 2 line 68 it states that Covid-19 vaccines were excluded. How was this done? Was there any information given to participants about which vaccines to consider – they may feel differently about different vaccines.

The Study was carried out before the COVID-19 vaccination, we have specified it better

In section 2.5 on statistical analysis, in the current text it is not clear what is being compared in the statistical tests or which factors were tested for inclusion in the logistic regression? It should also be explained how global vaccination coverage is defined.

All factors with p values lower than 0.25 were integrated into the multivariate logistic regression model

We specify the definition of the global vaccination coverage : »Eleven vaccines have become mandatory in 2018: : DTPcoq, MMR, HBV, meningococcus C, pneumococcus, Haemophilus [12]. A global VC was defined when these 11 mandatory vaccines were up to date

More detail is also needed in the results section:

Page 3 line 109: Knowledge is not included in Figure 1 as stated in the text.

Knowledge about vaccination had been added in the figure 1

Page 3 line 110-112: Were the differences mentioned between personal health and others health and effectiveness and safety significant? Include p value for these tests.

We added the p value : » Students' interest in vaccination was not different  for the personal health (mean 8.6; SD:2.03) than others health (mean 8.5; SD:2.30) (p=0.30), and they have more confidence in effectiveness (mean 8.0; SD (2.27) than safety (mean 7.7 (SD: 2.34); p<0.0001. »

Table 1: Include p value for multiple logistic regression. Men should also be included in the table as the reference category.

 We added men as the reference category

Minor comments

Abstract

Page 1 Line 11: Explain what abbreviation VC is the first time appears

We explain VC : vaccination coverage

Page 1 Line 16: What is meant by score confidence?

 We remove the terme of score

Page 1 Line 18: “the own’s health” change to “their own health”

OK

Page 1 Line 19-22: incomplete sentence “education about the general interest and mechanism of action of vaccines”

We complete the sentence

Page 1 Line 20-22: revise sentence to improve clarity.

 We revise the sentence

Introduction

Eg Page 1 line 26: “diseases cases” to “disease cases”

OK

Page 1 line 38: missing ] – check throughout as others missing as well.

OK

Page 1 line 38: Does national here refer to France. Consider introducing France earlier.

OK

Page 1 line 40: “little is known about young adults” change to “little is know about vaccination in young adults”.

OK

Page 1 line 41-2: include what the coverage is for these vaccines.

OK

Page 1 line 44: studies about what “are sparse among students”?

OK

Methods

Page 2 section 2.1: Include more details about the survey - How long did the survey take, was a reminder sent, was the survey in French?

The electronic questionnaire was send via the university mailing list of the students of Rouen-Normandy University with two e-mail reminder from the 7th to the 31st of January 2021’

Page 2 line 68: How were Covid-19 vaccines excluded? Was there any information given to participants about which vaccines to consider – they may feel differently about different vaccines eg measles compared to HPV.

There was one question per vaccine to collect vaccination status, there was no question on COVID-19 vaccination status but only on vaccination intention because the study took place before COVID-19 vaccination among young people in France

Page 2 line 81: In the introduction it states that HPV vaccines is now offered to 11 year old males why was the question about HPV VC only asked of females?

We pointed out that HPV vaccination has been offered since 2020 for boys aged 11 to 19, so it is too recent for the students in our studyPage 2 line 87: it is not clear what the global VC refers to. Is this a question in the survey?

Page 2 line 88: expand what DTPcoq is

 We modified DTPcoq by DTaP/IPV

Page 2 line 93: “student test” change to “student t-test”

OK

Page 2 line 93: “qualitative variables” I think should be discrete variables

OK

Page 2 section 2.5: it is not clear what is being compared in the statistical tests. Which factors were tested for inclusion in the logistic regression?

All factors with p values lower than 0.25 were integrated into the multivariate logistic regression model

Page 2 section 2.5: How is global vaccination coverage defined? Is it answering yes to having all listed vaccines and boosters?

 Yes « Eleven vaccines have become mandatory in 2018: : DTPcoq, MMR, HBV, meningococcus C, pneumococcus, Haemophilus [12]. A global VC was defined when these 11 mandatory vaccines were up to date »

Results

Page 3 line 100: what is consent rate?

We added : 10.3% of participation rate

Page 3 line 100: “mean of age” change to “mean age”

OK

Page 3 line 101: “with a mean of age of 20.3 100 (SD=1.9), 71.4% of women and 38.8% of healthcare students.” change to “with a mean of age of 20.3 100 (SD=1.9). Overall 71.4% of participants were women and 38.8% were healthcare students.”

OK

Page 3 line 110-112: Include the values in the text eg “Students' interest in vaccination is higher for personal health (mean 8.6, SD 2.03)”.

OK

Figure 1: remove “the” from all axis categories eg “confidence in efficacy” change to “confidence in efficacy

 OK

Discussion

Page 5 line 148: “cursus”should this be “courses”?

OK

Reviewer 2 Report

The authors present a cross-sectional survey from one university in France on perceptions about vaccination. Data were collected prior to the availability of a COVID-19 vaccination; thus, the authors ask students about their intention to be vaccinated against the virus. The paper is generally well-written and addresses an important topic. However, there are several limitations to the study.

  1. The survey was conducted at a single university. Although Rouen-Normandy has a large student population (> 30,000), it is not clear if this sample is representative of any population (university students, young adults, etc.) in France.
  2. Even if the student body at Rouen-Normandy is representative of college students or young adults in France, it is not clear that the sample who completed the survey are representative of the student body as a whole. The response rate was low (~ 10%) and the authors do not provide information about whether they attempted to increase response rates. As written, this appears to be a one-shot, cross-sectional survey with a low response rate.
  3. I read the methods several times and I wasn’t sure if the flu vaccine was required to be in the “global VC” category. It is also not clear whether HPV vaccine resulted in differences among males and females on the percent who were in the global VC category. Essentially, more was required of women compared to men to be considered fully vaccinated.
  4. The data are fairly dated. The COVID-19 vaccines have been available for quite some time, so providing only data about the student’s prediction of whether they will be vaccinated for the virus is a significant limitation. A follow-up survey of whether students received a COVID-19 vaccine would significantly improve the utility of the study are unadjusted comparisons within participant characteristics. The fact that very small differences between groups all lead to extremely small p-values suggests that the authors should consider whether the effect sizes are practically meaningful. For example, there was a 1.1-point difference on a 10-point scale for “Knowledge About Vaccination” which had a p-value of <0.0001. None of the mean differences seemed large. The odds ratios from the multivariate model were mostly in the “very small” range of magnitude.
  5. The authors noted that factors with univariate p-values below 0.25 were included in the multivariate model. That would mean all variables based on Table 1 except gender, correct? If that is the case, there should also be an effect size criterion given that with a large sample size even trivial differences can be statistically significant.
  6. Given the large sample size, stratifying on other factors, such as gender or age, may be enlightening.
  7. Figure 1 suggests very small differences between groups with very small p-values. Again, a focus on the magnitude of mean differences is needed. Also, it is not clear whether the error bars are SD or SE.
  8. In figure 2, fewer people received the flu vaccine than were categorized as being globally vaccinated. I assume that means that the flu vaccine did not count toward being globally vaccinated.
  9. Many universities require vaccination for admission. It would be helpful for the authors to discuss any vaccination policy in the region or at the university.

Minor points: Please provide sample size and recruitment rates in the abstract.

Author Response

The authors present a cross-sectional survey from one university in France on perceptions about vaccination. Data were collected prior to the availability of a COVID-19 vaccination; thus, the authors ask students about their intention to be vaccinated against the virus. The paper is generally well-written and addresses an important topic. However, there are several limitations to the study.

We thank the reviewer for their helpful comments to improve our manuscript.

  1. The survey was conducted at a single university. Although Rouen-Normandy has a large student population (> 30,000), it is not clear if this sample is representative of any population (university students, young adults, etc.) in France.

We indicated in the limitation section »The student population in the city of Rouen, France, is made up of approximately 30,000 individuals, and the response rate in this study was around 10%, which could engender a response bias and may limit its generalizability and representativeness. The proportion of women is slighty higher than women students in France (60%). The proportion of HCS was higher in our study (39%) than in the University of Rouen-Normandy (15%); however, HCS and non-HCS were separately analyzed, which limited the selection bias. »

Even if the student body at Rouen-Normandy is representative of college students or young adults in France, it is not clear that the sample who completed the survey are representative of the student body as a whole. The response rate was low (~ 10%) and the authors do not provide information about whether they attempted to increase response rates. As written, this appears to be a one-shot, cross-sectional survey with a low response rate.

The electronic questionnaire was send via the university mailing list of the students of Rouen-Normandy University with two e-mail reminder from the 7th to the 31st of January 2021.

  1. I read the methods several times and I wasn’t sure if the flu vaccine was required to be in the “global VC” category. It is also not clear whether HPV vaccine resulted in differences among males and females on the percent who were in the global VC category. Essentially, more was required of women compared to men to be considered fully vaccinated.

Flu vacccine was not required in the global VC : ‘Eleven vaccines have become mandatory in 2018: DTaP/IPV, MMR, HBV, meningococcus C, pneumococcus, Haemophilus [12]. A overall VC was defined when these 11 mandatory vaccines were up to date.’

 We add this sentence « There is no difference of the global VC between men  (27.0%) and women (28.5%) p=0.22 »

  1. The data are fairly dated. The COVID-19 vaccines have been available for quite some time, so providing only data about the student’s prediction of whether they will be vaccinated for the virus is a significant limitation. A follow-up survey of whether students received a COVID-19 vaccine would significantly improve the utility of the study are unadjusted comparisons within participant characteristics. The fact that very small differences between groups all lead to extremely small p-values suggests that the authors should consider whether the effect sizes are practically meaningful. For example, there was a 1.1-point difference on a 10-point scale for “Knowledge About Vaccination” which had a p-value of <0.0001. None of the mean differences seemed large. The odds ratios from the multivariate model were mostly in the “very small” range of magnitude.

Globally students were vaccinated in 2021 against COVID-19 because vaccination is mandatory for students in health and 98% of French 18-29 years are vaccinated

1 point/10 is an important difference and the multivariate analysis allows us to highlight possible action levers,

The effectif size of the odds ratio were good :  beetween 0.4 and 0.5

  1. The authors noted that factors with univariate p-values below 0.25 were included in the multivariate model. That would mean all variables based on Table 1 except gender, correct? If that is the case, there should also be an effect size criterion given that with a large sample size even trivial differences can be statistically significant.

The gender was also included in the multivariate analysis (p=0.22 in invariate analysis)

  1. Given the large sample size, stratifying on other factors, such as gender or age, may be enlightening.

The tests of interactions with gender are p>0.05 so it is not necessary to do a stratified analysis

  1. Figure 1 suggests very small differences between groups with very small p-values. Again, a focus on the magnitude of mean differences is needed. Also, it is not clear whether the error bars are SD or SE.

The errors bars are SD

  1. In figure 2, fewer people received the flu vaccine than were categorized as being globally vaccinated. I assume that means that the flu vaccine did not count toward being globally vaccinated.

Flu vaccination is not included in the global VC

  1. Many universities require vaccination for admission. It would be helpful for the authors to discuss any vaccination policy in the region or at the university.

Hepatitis B was mandatory and flu recommended for health students. We added this information in the discussion section

 Minor points: Please provide sample size and recruitment rates in the abstract.

We added this information

Reviewer 3 Report

An important issue in cross sectional studies is about method of sampling. The convenience sampling is not appropriate. Also data gathering with online questionnaire is not representative, because there are some students without access to internet or mailing list.

It is better you survey the parents vaccination status, because students behaviour is dependent to parents behaviour.

Author Response

An important issue in cross sectional studies is about method of sampling. The convenience sampling is not appropriate. Also data gathering with online questionnaire is not representative, because there are some students without access to internet or mailing list.

The methods of sampling does not present a selection bias by the online collection :  All students have an university email that has been used for a mailing list and 99% have an Internet acces (smartphone or personal computer)

We added the sentence « The electronic questionnaire was send via the students' university e-mail list of the students of Rouen-Normandy University »

It is better you survey the parents vaccination status, because students behaviour is dependent to parents behaviour.

It is relevant to study the students vaccinaion beacause the students are also future adults and future parents and healthcare students are future healthcare workers and will vaccinate and counsel their patients